# Mitigating Temporal Misalignment by Discarding Outdated Facts

**Michael J.Q. Zhang** and **Eunsol Choi**
Department of Computer Science
The University of Texas at Austin
{mjqzhang, eunsol}@utexas.edu

## Abstract

While large language models are able to retain vast amounts of world knowledge seen during pretraining, such knowledge is prone to going out of date and is nontrivial to update. Furthermore, these models are often used under temporal misalignment, tasked with answering questions about the present, despite having only been trained on data collected in the past. To mitigate the effects of temporal misalignment, we propose *fact duration prediction*: the task of predicting how long a given fact will remain true. In our experiments, we demonstrate that identifying which facts are prone to rapid change can help models avoid reciting outdated information and determine which predictions require seeking out up-to-date knowledge sources. We also show how modeling fact duration improves calibration for knowledge-intensive tasks, such as open-retrieval question answering, under temporal misalignment, by discarding volatile facts. Our data and code are released publicly at https://github.com/mikejqzhang/mitigating_misalignment.

## 1 Introduction

A core challenge in deploying NLP systems lies in managing *temporal misalignment*, where a model that is trained on data collected in the past is evaluated on data from the present (Lazaridou et al., 2021). Temporal misalignment causes performance degradation in a variety of NLP tasks (Luu et al., 2021; Dhingra et al., 2022; Zhang and Choi, 2021). This is particularly true for knowledge-intensive tasks, such as open-retrieval question answering (QA) (Chen et al., 2017), where models must make predictions based on world knowledge which can rapidly change. Furthermore, such issues are only exacerbated as the paradigm for creating NLP systems continues to shift toward relying on large pretrained models (Zhang et al., 2022; Chowdhery et al., 2022) that are prohibitively expensive to retrain and prone to reciting outdated facts.

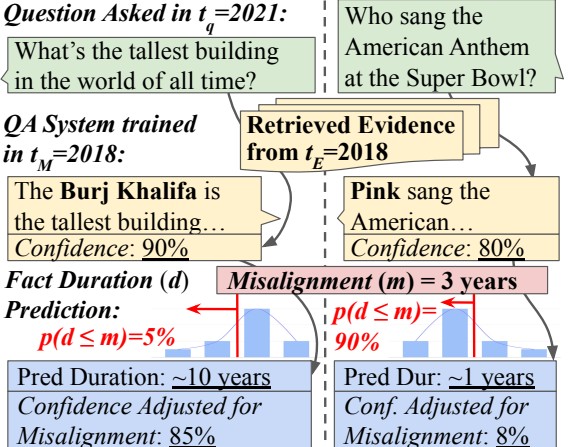

Figure 1: We depict the critical timestamps at play in open-retrieval QA systems. In the example on the left, the temporal misalignment between when the system was trained and evaluated has no affect on the answer. On the right, the answer has changed, causing the system to output an outdated answer with high confidence. To account for this, we apply our fact duration prediction system to adjust the system's confidence accordingly.

Prior work has attempted to address these issues by updating the knowledge stored within the parameters of an existing pretrained model (Cao et al., 2021; Mitchell et al., 2022; Onoe et al., 2023). Another line of work has proposed using retrieval-based systems, which utilize a nonparametric corpus of facts that can be updated over time (Karpukhin et al., 2020; Guu et al., 2020; Lewis et al., 2021). Both methods, however, are incomplete solutions as they rely on an oracle to identify which facts need to be updated and to continuously curate a corpus of up-to-date facts.

Given the difficulty of keeping existing models up-to-date, we propose an alternative solution where we *abstain* from presenting facts that we *predict* are out of date.[1] To accomplish this, we

---

[1]Some modern language model assistants (e.g., ChatGPT) have demonstrated a similar ability to abstain from responding to questions with temporally-dependent answers. We compare our methods against such system in Section 5.3.

introduce **fact duration prediction**, the task of predicting how frequently a given fact changes, and establish several classification and regression-based baselines. We also explore large-scale sources of distant supervision for our task, including fact durations extracted from temporal knowledge bases (Chen et al., 2021b) and duration-related news text (Yang et al., 2020). We provide rich discussion on this challenging task, exploring the relationship between fact duration prediction and temporal commonsense (Zhou et al., 2020).

We provide two sets of evaluations for our fact duration prediction systems. First, as intrinsic evaluations, we report how close our systems' duration estimates are to ground truth labels. We find that models that are trained with only distant supervision can predict the duration of 65% of temporally dependent facts from real search queries in NaturalQuestions (Kwiatkowski et al., 2019) to within 3 years, compared to 11% from a simple average-duration baseline. Second, in extrinsic evaluations, we measure how systems' duration estimates can improve an open-retrieval QA system's predictions under temporal misalignment. We mainly focus on improving calibration (as depicted in Figure 1). Our approach can reduce expected calibration error by 50-60% over using system confidence alone on two QA systems (Roberts et al., 2020; Karpukhin et al., 2020) on the SituatedQA dataset (Zhang and Choi, 2021).

Lastly, we also explore other ways of applying our fact duration systems in QA. We experiment with adaptive inference in ensembled open/closed-book QA systems, using duration prediction to decide when retrieval is necessary due to temporal misalignment. We also apply fact duration prediction in a scenario where retrieval is performed over heterogeneous corpus containing both outdated and recent articles, and systems must weigh the relevance of an article against its recency. In summation, we present the first focused study on mitigating temporal misalignment in QA through estimating the duration of facts.

## 2 Settings

We aim to address temporal misalignment (Luu et al., 2021) in knowledge-intensive tasks, such as open-retrieval QA. Figure 1 illustrates our setting. We assume a QA model that is developed in the past and is evaluated on a query from a later date. This system suffers from temporal misalignment, return-

| Eval. Year | Model | EM ↑ | AUR-OC ↑ | ECE ↓ | RC@55 ($|\Delta| \downarrow$) |
|---|---|---|---|---|---|
| 2018 | ① T5 | 36.0 | 0.82 | 0.13 | 55.0 (0.0) |
| | ② DPR | 37.9 | 0.74 | 0.22 | 55.0 (0.0) |
| 2021 | ① T5 | 17.4 | 0.77 | 0.26 | 27.2 (27.8) |
| | ② DPR | 17.1 | 0.63 | 0.43 | 22.4 (32.6) |
| *Adjusting Confidence With Oracle Misalignment Info.* | | | | | |
| 2021 | ① T5 | 17.4 | 0.75 | 0.12 | 47.8 (7.2) |
| | ② DPR | 17.1 | 0.71 | 0.17 | 38.7 (16.3) |

Table 1: NQ-Open QA performance evaluated on answers from 2018 (from NQ-Open) and 2021 (from SituatedQA). Confidence estimates are taken from calibration models that have been trained for each QA system. All models are trained on 2018 answers from NQ-Open. On the bottom, we compare against an oracle system which zeros the confidence of predictions whose answers have changed between 2018 and 2021.

ing outdated answers for some questions whose answer has changed in the meantime.

Table 1 reports the QA performance of existing systems on SituatedQA (Zhang and Choi, 2021), a subset of questions from NQ-Open (Kwiatkowski et al., 2019; Lee et al., 2019) that has been re-annotated with the correct answer as of 2021. In this dataset, 48% of questions are updated within the temporal gap (2018 to 2021). We can see that the current models, without considering temporal misalignment, experience performance degradation on both answer accuracy (EM) and calibration.[2]

In this table, we also explore using an oracle that identifies which answers have changed and zeroes the QA system's confidence in such predictions. While this does not change the system's accuracy, it helps models identify incorrect predictions, improving calibration metrics across the board. In real-world scenarios, however, we do not know which facts are outdated. Thus, in this work we build a fact duration model which predicts facts that are likely outdated and use it to adjust the confidence of the QA model. We introduce our fact duration prediction and QA settings in detail below.

### 2.1 Fact Duration Prediction

We define the fact duration prediction task as follows: given a fact $f$, systems must predict its duration $d$, the amount of time that the fact remained true for. We consider datasets that represent facts in a variety of formats: QA pairs, statements,

---

[2]QA and calibration model details can be found in Section 5.1 and calibration metrics are explained in Section 2.2.

knowledge-base relations. For modeling purposes, we convert all facts to statements. For example, the fact $f =$ *"The last Summer Olympic Games were held in Athens."* has a duration of $d = 4$ years.

**Error Metrics**    We evaluate fact duration systems by measuring error compared to the gold reference duration: **Year MAE** is the mean absolute error in their predictions in years and **Log-Sec MSE** is mean squared error in log-seconds.

## 2.2 QA under Temporal Misalignment

The open-retrieval QA task is defined as follows: Given a question $q^i$, a system must produce the corresponding answer $a$, possibly relying on retrieved knowledge from an evidence corpus $E$. When taking temporal misalignment into consideration, several critical timestamps can affect performance:

- **Model Training Date ($t_M$):** When the training data for $M$ was collected or annotated.
- **Evidence Date ($t_E$):** When $E$ was authored.[3]
- **Query Date ($t_q$):** When $q$ was asked.

For studying QA under temporal misalignment, we further specify that systems must produce appropriate answer at the time of the query $a_{t_q}$. For example, the question $q =$ *"Where are the next Summer Olympics?"* asked at $t_q = 2006$ has answer $a_{2006} =$ *"Beijing"*. We define the magnitude of the temporal **misalignment** ($m$) to be the amount of time between a a model's training date and the query date ($m = t_M - t_q$). We will compare this with the duration of the fact being asked $d = f(q, a_{t_q})$. If $m > d$, we should **lower** the confidence of the model on this question.

For simplicity, we do not take an answer's start date into account. Ideally, determining whether a given QA pair $(q, a)$ has gone out of date should also consider the answer's start date ($t_s$) and a model's training date ($t_m$), and confidence can be lowered if $t_s + d < t_m + m$. While we expect this approximation to have less of an impact when settings where the misalignment period is small with respect to the distribution of durations, we perform error analysis on examples where considering start date hurts performance in Appendix C.

---

[3]We primarily study settings where the evidence corpus does not change between training and inference ($t_E = t_M$), but also explore the effects of updating the evidence corpus ($t_E = t_q$) in Section 6. Training corpora and evidence corpora can contain documents authored over a span of time. For $t_M$, we use the date its *latest* document was authored. For, $t_E$, we studying using Wikipedia as our evidence corpus, and therefore all evidence is up-to-date as of $t_E$.

**Calibration Metrics**    Even without temporal misalignment, models will not always know the correct answer. Well calibrated model predictions, however, allow us to identify low-confidence predictions and avoid presenting users with incorrect information (Kamath et al., 2020). Under temporal misalignment, calibration further requires identifying which predictions should receive reduced confidence because the answer has likely changed. We consider following calibration metrics:

- **AUROC:** Area under the ROC curve evaluates a calibration system's performance at classifying correct and incorrect predictions over all possible confidence thresholds (Tran et al., 2022).
- **Expected Calibration Error (ECE):** Computed by ordering predictions by estimated confidence then partitioning into 10 equally sized buckets. ECE is then macro-averaged absolute error each bucket's average confidence and accuracy.
- **Risk Control (RC@XX):** Uncertainty estimates are often used for selective-prediction, where models withhold low-confidence predictions below some threshold ($< \tau$), where $\tau$ is set to achieve a target accuracy (XX%) on some evaluation set. We measure how well $\tau$ generalizes to a new dataset (Angelopoulos et al., 2022). To compute RC@XX, we set $\tau$ based on predictions from $t_M$, then compute the accuracy on predictions from $t_q$ with confidence $\geq \tau$. In the ideal case, the difference ($|\Delta|$) between RC@XX and XX should be zero.

## 3 Data

We first describe the datasets used for evaluation, split by task. We then describe our two large-scale sources for distant supervision. Appendix B contains further prepossessing details and examples.

### 3.1 Evaluation Datasets

**QA under Misalignment**    Our primary evaluations are on SituatedQA (Zhang and Choi, 2021), a dataset of questions from NQ-Open (Kwiatkowski et al., 2019) with temporally or geographically dependent answers. We use the temporally-dependent subset, where each question has been annotated with a brief timeline of answers that includes the correct answer as of 2021, the prior answer, and the dates when each answer started to be true. We evaluate misalignment between $t_M = 2018$ and $t_q = 2021$ using the answers from NQ-Open for $a_{2018}$ and answers from SituatedQA as $a_{2021}$.

While several recent works have proposed new

| QA Calibration Under Temporal Misalignment | Total (Ch. / Unch. between $t_M = 2018$ and $t_q = 2021$) |
|---|---|
| SituatedQA | 322 (157 / 165) |

| Duration Prediction | # Train / Dev / Test |
|---|---|
| SituatedQA | — / 377 / 322 |
| MC-TACO (Duration) | — / 1,075 / 2,899 |
| TimeQA | 11,708 / 2,492 / 2,461 |
| TimePre | 24,089 / 5,686 / — |

Table 2: Dataset statistics for our QA misalignment calibration and duration prediction tasks. We report the number of examples used in our QA calibration experiments along with how many examples have answers that have changed/unchanged between 2018 and 2021.

datasets for studying temporal shifts in QA (Kasai et al., 2022; Livska et al., 2022), these works focus on questions about new events, where answers do not necessarily change (e.g., "How much was the deal between Elon Musk and Twitter worth?"). We do not study such shifts in the input distribution over time. We, instead, study methods for managing the shift in the output distribution (i.e., answers changing over time). Adjusting model confidence due to changes in input distribution has been explored (Kamath et al., 2020); however, to the best of our knowledge, this is the first work on calibrating over shifts in output distribution in QA.

**Fact Duration** Following suit with the QA evaluations above, we also evaluate fact duration prediction on SituatedQA. To generate fact-duration pairs, we use the annotated previous answer as of 2021, converting the question/answer pair into statement using an existing T5-based (Raffel et al., 2020) conversion model (Chen et al., 2021a). We then use distance between the 2021 and previous answer's start date as the fact's duration, $d$.

**Temporal Commonsense** Temporal commonsense focuses on inferences about generic events (e.g., identifying that glaciers move over centuries and a college tours last hours). In contrast, fact duration prediction requires making inferences about specific entities. For instance, determining the duration of an answer to a question like *"Who does Lebron James plays for?"* requires entity knowledge to determine that *Lebron James* is a basketball player and commonsense knowledge to determine that basketball players often change teams every few years. Previous work (Onoe et al., 2021) has demonstrated the non-trivial nature of combining entity-specific and commonsense knowledge.

Due to the differences described above, we do not use temporal commonsense datasets for evaluating fact duration prediction. We, however, still evaluate on them to explore how these tasks compare. In particular, we evaluate our fact duration systems on the event duration subset of MCTACO (Zhou et al., 2019). Each MCTACO example consists of a multiple-choice question about the duration of some event in a context sentence, which we convert into duration statements. We evalute using the metrics proposed by the original authors. Following Yang et al. (2020), we select all multiple choice options whose duration falls within a tuned threshold of the predicted duration. **EM** measures accuracy, evaluating whether the gold and predicted answer sets exactly match. **F1** measures the average F1 between the gold and predicted answer sets.

### 3.2 Distant Supervision Sources

**Temporal Knowledge Bases** have been used in numerous prior works for studying how facts change over time. TimeQA (Chen et al., 2021b) is one such work that curates a dataset of 70 different temporally-dependent relations from Wikidata and uses handcrafted templates to convert into decontextualized QA pairs, where the question specifies a time period. To convert this dataset into fact-duration pairs $(f, d)$, we first convert their QA pairs into a factual statements by removing the date and using a QA-to-statement conversion model (Chen et al., 2021a). We then determine the duration of each facts to be the length of time between the start date of one answer to the question and the next.

**News Text** contains a vast array of facts and rich temporal information. Time-Aware Pretraining dataset (TimePre) (Yang et al., 2020) curates such texts from CNN and Daily Mail news articles using regular expressions to match for duration-specifying phrases (e.g., "Crystal Palace goalkeeper Julian Speroni has ended the uncertainty over his future by signing a new *12 month* contract to stay at Selhurst Park."). Pretraining on this dataset has previously been shown to improve performance on temporal commonsense tasks.

### 3.3 Dataset Summary

Table 2 reports data statistics and Figure 2 presents the distribution of durations from each dataset. While most facts in SituatedQA and TimeQA change over the course of months to decades, facts in MCTACO and TimePre cover a wider range.

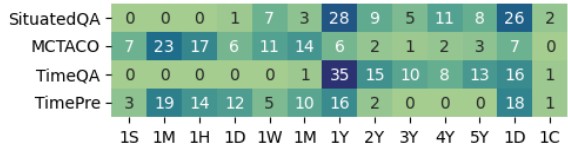

Figure 2: Duration statistics on each dataset's development set. Columns represent different duration classes used by our classification model, with units abbreviated as **S**econds, **M**inutes, **D**ays, **W**eeks, **M**onths, **Y**ears, **D**ecades, and **C**enturies. Cells contain the % of examples in each dataset in the column's duration class.

## 4 Fact Duration Prediction

### 4.1 Comparison Systems

Here, we describe our fact duration prediction systems. We include two simple lowerbound baselines: **Random** samples a duration and **Average** uses the average duration from each dataset.

Following prior work on temporal common sense reasoning (Yang et al., 2020), we develop BERT-based (Devlin et al., 2018) models.[4] We frame fact duration prediction as cloze questions to more closely match the system's pretraining objective (Schick and Schütze, 2020). To this end, we append ", lasting [MASK][MASK]" onto each fact, eliciting the model to fill in the masked tokens with a duration. We use two mask tokens as typically duration information requires at least two tokens, one for value and another for unit. For our TimePre and MCTACO datasets, we similarly replace the target durations with two mask tokens. Table 8 in Appendix B contains examples. Predictions are made by averaging the encoded representations of the two "[MASK]" tokens, then using this representation as an input to a single hidden layer network. Using this same representation, we train two models with regression-based and classification-based learning objectives described below.

**Classification Model** frames the task as a 13-way classification task where each class corresponds to a duration in Figure 2.[5] We train using cross entropy loss, selecting the closest duration class as the pseudo-gold label for each fact. Because this model can only predict a limited set of durations, we report its upperbound by always selecting the class closest to the gold duration.

---

[4] We explore using other pretrained models in Appendix C.
[5] We select these duration classes based on frequency across all datasets.

**Regression Model** uses the mean squared error loss in units of log seconds, where the output from the hidden layer predicts a scalar value.

### 4.2 Results

We experiment with training on TimeQA and TimePre individually and on the union of both datasets. Figure 3 reports duration prediction and temporal commonsense performance. Overall, we find that our trained systems outperform simple random and average baselines on SituatedQA. This is indicative of strong generalizability from our distantly-supervised fact-duration systems, even when baselines benefit from access to the gold label distribution. We also provide a histogram of errors from our systems in Figure 3 where we can see that over 60% of our classification-based system's predictions are within 3 years of the gold duration, while predicting the exact duration remains challenging. Below, we reflect on the impact of our different modeling choices and research questions.

**Regression vs. Classification** Regression-based models tend to outperform their classification-based counterparts. The instances where this is not true can be attributed to an insufficient amount of training data. In Figure 3, we can see the different types of errors each model makes. The classification system predicts duration within 1 year more frequently, but the regression system predicts duration within 4 years more frequently.

**Supervision from KB vs. News Text** We find that training on temporal knowledge-base relations (TimeQA) alone vastly outperforms training on news text (TimePre) alone for fact-duration prediction; however, the opposite is true when comparing performance on temporal commonsense (MCTACO). Training on both datasets tends to improve our regression-based system, but yields mixed results for our classification-based system. We hypothesize that the closeness in label distribution (see Figure 2) between the training and evaluation sets impacts the performance significantly.

**Fact Duration vs. Temporal Commonsense** While fact duration prediction and temporal commonsense are conceptually related, we find that strong performance on either task does not necessarily transfer to the other. As discussed above, this can be attributed to differences in label distributions; however, label distribution also serves as a proxy variable for the type of information being

| Model | Training Data | SituatedQA | | TimeQA | | MCTACO | |
|---|---|---|---|---|---|---|---|
| | | MSE (LS, ↓) | MAE (Y, ↓) | MSE (LS, ↓) | MAE (Y, ↓) | EM (↑) | F1 (↑) |
| Random | — | 7.14 | 13.44 | 2.22 | 6.96 | 20.3 | 38.1 |
| Average | — | 5.42 | 10.61 | 1.65 | 5.41 | 23.8 | 41.3 |
| Classification | Oracle | 0.28 | 4.18 | 0.12 | 1.66 | 8.0 | 37.2 |
| | TimeQA | **4.20** | 8.45 | **1.55** | 4.80 | 20.3 | 41.3 |
| | TimePre | 40.51 | 8.77 | 9.36 | 7.85 | **28.3** | **57.7** |
| | Time(QA+Pre) | 6.28 | **8.37** | 1.70 | **4.66** | 28.3 | 57.2 |
| Regression | TimeQA | 3.75 | 8.40 | 0.97 | 4.22 | 21.2 | 41.0 |
| | TimePre | 38.48 | 8.99 | 32.10 | 5.70 | **33.1** | **57.8** |
| | Time(QA+Pre) | **3.58** | **8.15** | **0.97** | 4.19 | 31.8 | 57.8 |

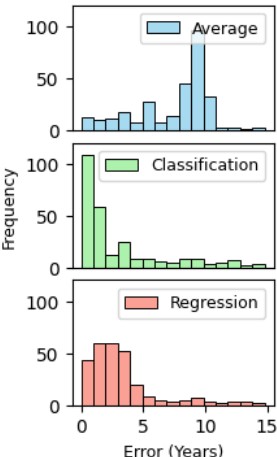

Figure 3: Fact Duration Prediction Results. On the left, we report our full results, with performance split by model type and training data. Performance on SituatedQA and TimeQA are given as the mean average error in years (Y) and mean squared error in years in log-seconds (LS), the same as the regression system training loss. On the right, we depict error histograms evaluated on SituatedQA, with systems trained on TimeQA.

queried for in either task. Commonsense knowledge primarily differentiates events that take place over different orders of magnitude of time (e.g., seconds versus years). Differentiating whether an event takes place over finer-grained ranges (e.g., one versus two years), however, cannot be resolved with commonsense knowledge alone, and further require fact retrieval. We find that NQ contains queries for facts that change over a smaller range of durations (between 1-10 years), and, therefore, commonsense knowledge alone is insufficient.

## 5 Calibrating QA under Temporal Misalignment

Here, we return our motivating use-case of using fact duration prediction to calibrate open-retrieval QA systems under temporal misalignment. We assume an access to base calibration system, $c(q,a) \in [0,1]$ that has the same training date as the QA system its calibrating. We then use fact duration to generate **misalignment-aware confidence score** $c_m$ through simple post-hoc augmentation, scaling **down** the original confidence score by a discount factor based on the degree of misalignment and the predicted fact duration. We compute this factor differently for each of our fact duration systems.

- **Classification:** Here, the system's output is a probability distribution over different duration classes, $p(d|q,a)$. We set the discount factor to

be the CDF of this distribution evaluated at $m$:[6] $c_m = c(q,a) \sum_{d \leq m} P(d|q,a)$.
- **Regression:** Here, the output is a single predicted duration $d$. We set the discount factor to the binary value indicating whether or not the misalignment period has exceeded the predicted duration: $c_m = c(q,a)\mathbb{1}\{d < m\}$

As classification systems predict a distribution over fact durations, we are able to use the CDF of this distribution to make granular adjustments to confidence over time. In contrast, our regression systems predict a single value for fact duration, and confidence adjustments over time are abrupt, leaving confidence unchanged or setting it to zero.

### 5.1 Models

**Base QA and Calibration Systems** We experiment with three QA systems throughout our study:
- ① T5: We use T5-large (Roberts et al., 2020) which has been trained with salient-span-masking and closed-book QA on NQ-Open.
- ② DPR ($t_e$=2018): We use DPR (Karpukhin et al., 2020), an open-book system which retrieves passages from a $t_e$ = 2018 Wikipedia snapshot and is also trained on NQ-Open.
- Ⓝ DPR ($t_e$=2021): We use the same model as ②, but swap the retrieval corpus with an updated Wikipedia snapshot that matches query times-

---

[6]We experiment with using our classification-based system's predicted class or the expected duration as the duration for adjusting confidence, but both methods underperform compared to using the system's CDF. We include these results in Appendix C.

| QA Model | 2018 → 2021 EM | Dur. Model | AUROC ↑ | ECE ↓ | RC@55 (\|Δ\|↓) | Avg Conf % Δ |
|---|---|---|---|---|---|---|
| ① T5 | 36.0 → 17.4 | N / A | 0.766 | 0.265 | 27.2 (27.8) | 0.0 |
| | | Oracle | 0.749 | 0.116 | 47.8 (7.2) | -25.8 |
| | | Regression | 0.709 | 0.185 | 32.4 (**22.6**) | -23.2 |
| | | Classification | **0.765** | **0.131** | 29.3 (25.7) | -15.1 |
| ② DPR ($t_e = 2018$) | 37.9 → 17.1 | N / A | 0.629 | 0.433 | 22.4 (32.6) | 0.0 |
| | | Oracle | 0.708 | 0.172 | 38.7 (16.3) | -36.9 |
| | | Regression | 0.601 | 0.268 | 26.1 (28.9) | -34.0 |
| | | Classification | **0.654** | **0.235** | 43.5 (**11.5**) | -20.9 |
| Ⓝ DPR ($t_e = 2021$) | —→ 19.6 | N / A | 0.636 | 0.370 | 25.4 (29.6) | -3.8 |

Table 3: Results for calibrating QA under temporal misalignment on SituatedQA. All systems' training dates are 2018 and evaluation dates are 2021. We report each system's EM accuracy, evaluated against the answers from 2018 and 2021. We also report how much model confidence changes on average (Avg Conf % Δ) with each adjustment method (for DPR with $t_e = 2021$ we compare average confidence against using $t_e = 2018$).

| Model | Adj. | AUROC | ECE | RC@55 (\|Δ\|) |
|---|---|---|---|---|
| ① T5 | Uniform | 0.757 | 0.180 | 30.5 (24.5) |
| | Per-Ex | **0.765** | **0.131** | **29.3 (25.7)** |
| ② DPR | Uniform | 0.627 | 0.259 | 25.6 (29.4) |
| | Per-Ex | **0.654** | **0.235** | **43.5 (11.5)** |

Table 4: Ablating per-example calibration: We first adjust confidence **Per-Ex**ample, which is our full system. We then adjust confidence **Uniform**ly across all examples, such that the net decrease in confidence across the entire test set is equivalent.

tamp ($t_e = 2021$) following Zhang and Choi (2021), which showed partial success in returning up-to-date answers.

For each QA system, we train a calibrator that predicts the correctness of the QA system's answer. We follow Zhang et al. (2021) for the design and input features to calibrator, using the model's predicted likelihood and encoded representations of the input (details in Appendix A).

**Fact Duration Systems** For both our regression and classification based models, we use the systems trained over both with TimeQA. We also include results using an oracle fact duration system, which zeroes the confidence for all questions that have been updated since the training date.

## 5.2 Results

Table 3 reports the results from our calibration experiments. Both QA models suffer from temporal degradation, and zero-ing out the confidence of outdated facts with oracle information improves the calibration performance. Using model prediction durations shows similar gains. Both re-

gression and classification duration predictions lower the confidence of models, improving calibration metrics across the board. We find that our classification-based model consistently outperforms our regression-based model on our calibration task, despite the opposite being true for our fact-duration evaluations. We attribute this behavior to our classification-based system's error distribution, as it gets more examples correct to within 1 year (Figure 3). Classification-based system also hedge over different duration classes by predicting a distribution, which we use to compute the CDF.

**Retrieval-Based QA: Update or Adjust** In Table 3, we compare the performance of DPR with static and hot-swapped retrieval corpora from $t_e = 2018$ and $t_e = 2021$. While updating the retrieval corpus improves EM accuracy, adjusting model confidence using fact duration on a static corpus performs better on all calibration metrics. This suggests that, when users care about having accurate confidence estimates or seeing only high-confidence predictions, confidence adjustments can be more beneficial than swapping retrieval corpus.

**Ablations: Per-Example vs Uniform Adjustment** We compare our system, which adjusts confidence on a per-example basis, against one that uniformly decreases confidence by the same value $v$ across the entire test set: $c_m = \max(c(q, a) - v, 0)$. These ablations still depend on our fact-duration systems to determine the $v$ such that the total confidence over the entire test set is the same for both methods. Table 4 reports the results from this ablation study. We find that uniformly adjusting confidence improves ECE, which is expected given

the decrease in the QA systems EM accuracy after misalignment. We find, however, that our per-example adjustment methods outperform uniform confidence adjustments.

## 5.3 Comparisons against Prompted LLMs

While we primarily focus on calibrating fine-tuned QA models, recent work has also explored calibrating prompted large language models (LLMs) for QA (Si et al., 2022; Kadavath et al., 2022; Cole et al., 2023). Furthermore, recent general-purpose LLMs (e.g., ChatGPT) have demonstrated the ability to abstain from answering questions on the basis that their knowledge cutoff date is too far in the past; however, it is not publicly known how these systems exhibit this behavior.

In this experiment, we investigate one such system, GPT-4 (OpenAI, 2023), and its ability to abstain from answering questions with rapidly changing answers. We prompt GPT-4 to answer questions from SituatedQA, and find that it abstains from answering 86% of all questions (95% of questions whose answers have been updated between 2018 and 2021 and on 79% of examples that have not). Overall, this behavior suggests that GPT-4 **overestimates** how frequently it must to abstain from answering user queries. Furthermore, GPT-4 does not provide how frequently the answer is expected to change. Collectively, GPT-4's tendency to over-abstain and its lack of transparency limits its usefulness to users. In contrast, our approach provides users with an duration estimate indicating why a prediction may not be trustworthy. Further experimental details and example outputs are reported in Appendix D.

## 6 Beyond Calibration: Adaptive Inference

In this section, we explore using our misalignment-aware confidence scores to decide **how** to answer a question. Below, we motivate and describe two adaptive inference scenarios where systems may choose between two methods for answering a question using their fact duration predictions.

**Hybrid: Closed + Open** (①+Ⓝ): Besides the computational benefits from not always having to use retrieval, forgoing retrieval for popular questions can also improve answer accuracy (Mallen et al., 2022). We use our fact duration predictions to decide when retrieval is necessary: we first predict an answer using T5 and run our fact duration

| Inference Ensemble | EM | % |
|---|---|---|
| ① T5 | 17.4 | 0.0 |
| ② DPR $t_e = 2018$ | 17.1 | 0.0 |
| Ⓝ DPR $t_e = 2021$ | 19.6 | 100.0 |
| ① T5 / Ⓝ DPR $t_e = 2021$ | **20.5** | 45.7 |
| ② DPR $t_e = 2018$ / Ⓝ DPR $t_e = 2021$ | 19.3 | 45.0 |

Table 5: Adaptive inference for temporal misalignment results: we use our duration prediction to decide whether to use the prediction from model with newer corpus (DPR $t_e = 2021$). In this data (SituatedQA), 48.8% of examples requires up-to-date knowledge.

prediction system using this answer. We then use the CDF of the predicted duration distribution to determine whether it is at least 50% likely that the fact has changed: $\sum_{d \leq m} P(d|q, a) \geq 0.5$. If so, we then run retrieval with DPR using the updated corpus $t_e = 2021$ and present the predicted answer. We report our results in the first row of Table 5, which shows that this outperforms either system on its own, while running retrieval on less than half of all examples.

**Two Corpora: Relevancy vs. Recency** (②+Ⓝ): While most work for QA have focused on retrieving over Wikipedia, many questions require retrieving over other sources such as news or web text. One challenge in moving away from Wikipedia lies in managing *temporal heterogeneity* across different articles. Unlike Wikipedia, news and web articles are generally not maintained to stay current, requiring retrieval-based QA systems to identify out-of-date information in articles. Systems that retrieve such resources must consider the trade-off between the recency versus relevancy of an article. In these experiments, we experiment with using fact duration prediction as a method for weighing this trade-off in retrieval.

Our experimental setup is as follows: instead of computing misalignment from the model's training date, we compute relative to when the article was authored ($m = 3$ years for 2018 Wikipedia and $m = 0$ years for 2021 Wikipedia). After performing inference using both corpora, we re-rank answers according to their misalignment-adjusted confidence estimates. We report results in Table 5. We find that our method is able to recover comparable performance to always using up-to-date articles, while using it just under half the time.

## 7 Related Work

**Commonsense and Temporal Reasoning** Recent works have proposed forecasting bench-

marks (Zou et al., 2022; Jin et al., 2021a) related to our fact duration prediction task. While our task asks models to predict *when* a fact will change, these forecasting tasks ask *how* a fact will change. Qin et al. (2021) studies temporal commonsense reasoning in dialogues. Quantitative reasoning has been explored in other works as quantitative relations between nouns (Forbes and Choi, 2017; Bagherinezhad et al., 2016), distributions over quantitative attributes Elazar et al. (2019), and representing numbers in language models (Wallace et al., 2019).

**Calibration** Abstaining from providing a QA system's answers has been explored in several recent works. Chen et al. (2022) examines instances where knowledge conflicts exist between a model's memorized knowledge and retrieved documents. As the authors note, such instances often arise due to temporal misalignment. Prior work (Kamath et al., 2020; Zhang et al., 2021; Varshney and Baral, 2023) has explored abstaining from answering questions by predicting whether or not the test question comes from the same training distribution of the QA system. While fact duration also predicts a shift in distribution, fact duration focuses on predicting a shift in a question's *output* distribution of answers instead of a shift in *input* distribution of questions; therefore, these two systems are addressing orthogonal challenges in robustness to distribution shift and are complementary.

**Keeping Systems Up-to-Date** Several works have explored continuing pretraining to address temporal misalignment in pretrained models (Dhingra et al., 2022; Jin et al., 2021b). Other works have explored editing specific facts into models (Cao et al., 2021; Mitchell et al., 2022; Meng et al., 2022). These works, however, have only focused on synthetic settings and assume access to the updated facts. Furthermore, such systems have yet to be successfully applied to new benchmarks for measuring whether language models have acquired emergent information (Onoe et al., 2022; Padmanabhan et al., 2023). Recent works on retrieval-based QA systems have found improved adaptation when updated with up-to-date retrieval corpora (Izacard et al., 2022; Lazaridou et al., 2022).

## 8   Conclusion

We improve QA calibration under temporal misalignment by introducing the fact duration pre-

diction task, alongside several datasets and baseline systems for it. Future work may build upon this evaluation framework to further improve QA calibration under temporal misalignment. For instance, future work may examine modeling different classes distributions of fact duration distributions, like modeling whether a fact changes after a regular, periodic time interval.

## Limitations

We only evaluate temporal misalignment between 2018 and 2021, a three-year time difference, on a relatively small scale SituatedQA dataset (N=322). This is mainly due to a lack of benchmark that supports studying temporal misalignment. Exploring this in more diverse setup, including different languages, text domains, wide range of temporal gaps, would be fruitful direction for future work.

As is the case with all systems that attempt to faithfully relay world knowledge, treating model predictions as fact runs the risk of propagating misinformation. While the goal of our fact duration prediction systems is to prevent models from reciting outdated facts, it does not always succeed and facts may change earlier than expected. Even though a given fact may be expected to only change once every decade, an improbable outcome may occur and the fact changes after only a year. In such an event, our misalignment-aware calibration system may erroneously maintain high confidence in the outdated answer.

Furthermore, our system, as it stands, does not take the answer start date into account. Our system also can make errors due to changes in the typical duration of a given fact. For instance *"What's the world's tallest building?"* changes more frequently over time as the rate of technological advances also increases. We provide examples of such system errors in Appendix C.

## Acknowledgements

The work was partially supported by Google Research Award, a gift from HomeDepot, and a grant from UT Machine Learning Lab. This work was in part supported by Cisco Research. Any opinions, findings and conclusions, or recommendations expressed in this material are those of the authors and do not necessarily reflect the views of Cisco Research. The authors would like to thank the members of the UT Austin NLP community and Jordan Boyd-Graber for helpful discussions.

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

# A  Implementation Details

## A.1  QA Models

We use the T5[7] and DPR[8] checkpoints that have been finetuned on NQ-Open's training set from the transformers library model hub.[9] For DPR, we use the retrieval corpora from December 20, 2018 for $t_E = 2018$ and February 20, 2021 for $t_E = 2021$, following Zhang and Choi (2021).

---

[7]https://huggingface.co/google/t5-large-ssm-nqo
[8]https://huggingface.co/facebook/dpr-reader-single-nq-base
[9]https://huggingface.co/models

## A.2  Calibration Models

We implement our trained calibration systems using XGBoost (Chen and Guestrin, 2016) using the features for T5 and DPR outlined in Zhang et al. (2021). For T5, we concatenate (1) the averaged, encoded representations of the input question, and (2) the model likelihood. For DPR, we concatenate (1) the averaged, encoded representations of the input question and selected passage, (2) the averaged, encoded representations of the start and end tokens of the selected answer span, and (3) the likelihood of the answer span, computed as the product of the likelihoods of selecting the start index, end index, and passage index.

We train our systems on NQ-Open on a randomly sampled 60/40 training and development splits, following Zhang et al. (2021). We use a maximum depth of 10 and experiment with several values for the learning rate $\{0.01, 0.1, 0.2, 0.5\}$ and column sub-sampling ratio $\{0.0, 0.1, \ldots, 0.9\}$, which we keep the for sampling by tree, level, and node. We train with early stopping after 10 epochs without improvement and select the best performing system as evaluated on the development split.

## A.3  Fact Duration Prediction Models

We use BERT-base from the transformers library for all duration prediction baselines, trained with a batch size in $\{32, 64\}$ and learning rate in $\{1e-5, 5e-5\}$. We train until convergence and select the best checkpoint as determined by development set performance. Due to computational resource constraints, we do not further tune hyperparameters. All models are trained once and results reflect a single run. All experiments took were performed Quadro RTX 8000 gpus and required less than one week's worth of GPU hours.

# B  Datasets

We provide examples from each dataset and our pre-possessing pipeline in Table 8. We provide futher preprocessing details below.

## B.1  Fact Duration Dataset Preprocessing

In TimeQA, several examples have answers that are simply the empty string. We remove all such examples from our preprocessed dataset. In SituatedQA, several examples have answers that begin and end in the same year, without further annotation determining the exact number of days or months. We

| Model | Training Data | SituatedQA | | TimeQA | | MCTACO | |
|---|---|---|---|---|---|---|---|
| | | LS-MSE | Y-MAE | LS-MSE | Y-MAE | EM | F1 |
| Classification | TimeQA | 8.5 | 4.18 | 4.7 | 1.47 | 20.6 | 41.2 |
| | TA Pretrain | 9.0 | 55.33 | 6.0 | 22.05 | 28.9 | 53.8 |
| | TimeQA + TA Pretrain | 8.2 | 7.58 | 5.1 | 1.56 | 28.0 | 56.9 |
| Regression | TimeQA | 8.4 | 3.76 | 4.6 | 1.18 | 25.1 | 41.3 |
| | TA Pretrain | 11.6 | 44.56 | 20.6 | 27.59 | 33.1 | 58.3 |
| | TimeQA + TA Pretrain | 8.7 | 7.66 | 4.6 | 1.23 | 32.2 | 57.3 |

Table 6: Fact Duration Prediction Results using DeBERTa-v3-base instead of BERT-base. All other settings are the same as in Figure 3.

| QA Model | Misalingmnent Adj. | AUROC ↑ | ECE ↓ | RC@55 ($|\Delta|$ ↓) | Avg Conf % $\Delta$ | |
|---|---|---|---|---|---|---|
| ① T5 $36.0 \rightarrow 17.4$ | CDF | 0.765 | 0.131 | 29.3 | 25.7 | 15.1 |
| | Argmax | 0.762 | 0.249 | 28.1 | 26.9 | 2.6 |
| | Expectation | 0.708 | 0.169 | 37.5 | 17.5 | 35.0 |
| ② DPR ($t_e = 2018$) $37.9 \rightarrow 17.1$ | CDF | 0.654 | 0.235 | 43.5 | 11.5 | 20.9 |
| | Argmax | 0.650 | 0.403 | 23.8 | 31.2 | 3.9 |
| | Expectation | 0.641 | 0.177 | 38.2 | 16.8 | 49.2 |

Table 7: Additional calibration results on SituatedQA, comparing different methods of adjusting model confidence for temporal misalignment using the output of our classification-based fact-duration system. All other settings are the same as in Table 3.

simply assume that all such question answer pairs instances have a duration of 1 month.

## B.2 Temporal Commonsense Datasets Preprocessing

As we noted above, each MCTACO example consists of a multiple-choice question about the duration of some event in a provided context sentence. During prepossessing, we use the same question conversion model as above to transform each QA pair into a statement and prepend the context sentence onto each question. We use the metrics proposed by the original authors, and we select all multiple choice options whose duration falls within some absolute threshold of predicted duration, measured in log seconds (Yang et al., 2020). This threshold is selected based on development set performance.

## C Additional Results

**Different Pretrained Models for Fact Duration Prediction** In Table 6, we report our results from DeBERTa-v3-base (He et al., 2021) on our fact duration prediction system. We also experiment with using the large variants of both BERT and DeBERTa, but do not find substantial improvement.

**Adjusting Confidence with Expected Duration** In addition adjusting confidence using the CDF of the predicted duration distribution from our classification-based system, we also experiment with using the expected duration as our discounting factor. We incorporate this by zeroing the confidence estimate if the expected duration is exceeded by the degree of misalignment: $f(q, a) = \mathbb{1}\{\sum_{d \geq m} d \cdot P(d|q, a)\}$.

In Table 7, we report additional results on our calibration evaluation. We include calibration performance of our best performing fact duration models finetuned on SituatedQA: trained only on SituatedQA for our classification-based model and first trained on TimeQA + TA Pretrain for our regression-based model.

**Error Analysis** In Table 9, we highlight sampled errors from our fact duration system and discuss their causes and impact.

## D ChatGPT and GPT-4 outputs

Table 10 includes two examples of ChatGPT informing users that the answers to a given question may have changed. It, however, does not provide users with an estimate of how likely it has changed, or how often the answer is expected to change. This lack of a duration estimate results in lesser transparency and interpretatbility for users. To get results on SituatedQA, we prompt GPT-4 with the following system prompt (recommended by their documentation): "You are ChatGPT, a large language model trained by OpenAI. Answer as con-

| Dataset | Example |
|---------|---------|
| SituatedQA | **Q:** Who are the judges on Asia Got Talent? / **A:** Vanness Wu / **Start:** 2015, **End:** 2017
**MI:** Vanness Wu is the judge on Asia Got Talent , lasting [MASK] [MASK] . / **TD:** 2 Years |
| MCTACO | **Context:** About 30% of Ratners's profit already is derived from the U.S.
**Q:** How long did it take to make profit? / **A:** 3 Months
**MI:** About 30% of Ratners's profit already is derived from the U.S. It took [MASK] [MASK] to make profit. / **TD:** 3 Months |
| TimeQA | **Subj:** Patrick Burns (businessman) **Rel:** Lives in **Obj:** Oshawa, Ontario **Start:** 1856 **End:** 1878
**MI:** Patrick Burns (businessman) lived in Oshawa, Ontario , lasting [MASK] [MASK] . / **TD:** 22 years |
| TA-Pretrain | **MI:** Jorge Ramos has been the face of Univision's News broadcast for [MASK] [MASK] . **TD:** 24 Years |

Table 8: Fact duration prediction input examples. We standardize formats to predict target duration (**TD**) from the masked input (**MI**). The top row(s) in each cell represents the original data, and the bottom row shows our setting.

| Quesiton {2018 Answer / 2021 Answer} | {G / P} Dur |
|--------------------------------------|-------------|
| When did the last volcano erupt in Iceland? (2010 / March 19, 2021) | 7 Y / 10 Y |
| How many episodes of Touching Evil are there? (16 / 16) | 1 W / 1 Y |
| Who got the most passing yards in the NFL? (Peyton Manning / Drew Brees) | 3 Y / 1 Y |

Table 9: Example fact duration predictions errors from SituatedQA. We include the answers from $t_M = 2018$ and $t_q = 2021$, and the gold / predicted durations ({G / P} Dur). In the first example, we see that our system correctly identified volcanic eruptions as an infrequent event; however, our system misses that the answer has changed within the misalignment period because it doesn't have access to the answer start date. In the second example, our system fails no longer changes since the show ended. In the last example, our model underestimates the fact's duration, but still correctly identifies that the answer has changed.

cisely as possible. Knowledge cutoff: September 2021. Current date: May 12, 2023." We then present GPT-4 with the user's question from SituatedQA. We determine whether a systems abstains from a given prediction if it references its knowledge cutoff from the prompt `September 2021` or if it mentions `real-time` information in its generated output.

# E  Licensing

**Dataset Licensing**  TimeQA is licensed under the BSD 3-Clause License. SituatedQA and Wikipedia are both distributed under the CC BY-SA 4.0 license. MCTACO and TA-Pretrain are both unlicensed.

**Model Licensing**  The transformers library, BERT model, T5 model, and question conversion model are all licensed under the Apache License v2.0. DPR is licensed under the Attribution-NonCommercial 4.0 International license. Our use

---

**User:** Who is the current US president?
**ChatGPT:** As an AI language model, I don't have real-time information or the ability to browse the internet. My knowledge was last updated in September 2021, and at that time, the President of the United States was Joseph R. Biden Jr. However, please note that the political landscape can change, and there may have been a new election or other developments since then. To obtain the most accurate and up-to-date information, I recommend checking reliable sources such as official government websites or reputable news outlets.

---

**User:** Who won the last NBA Champtionship?
**ChatGPT:** As an AI language model, I don't have real-time data, and my knowledge was last updated in September 2021. Therefore, I cannot provide information on the winner of the most recent NBA Championship. However, as of my last update, the Milwaukee Bucks were the reigning NBA Champions, having won the championship in the 2020-2021 season. I recommend checking reliable news sources or conducting a quick online search to find the current champion.

---

Table 10: Example ChatGPT outputs to two temporally dependent questions with different answer durations. Predictions are take from the May 12, 2023 ChatGPT version.

of all these resources are in compliance with their licenses.

# F  Ethical Considerations

We do not collect any data in this paper. While we do generate a dataset, we visually inspect generated examples and do not find any cases of harmful or offensive content. The existing datasets used in this work have previously been vetted by their authors prior work. We also note that our work is only applied to English datasets and models.