# OpenReview forum: "Mitigating Temporal Misalignment by Discarding Outdated Facts"
_EMNLP/2023/Conference — EMNLP 2023 Main_

### Official Review · Reviewer_TZQm · 2023-08-04

**Soundness:** 4

**Excitement:**

4: Strong: This paper deepens the understanding of some phenomenon or lowers the barriers to an existing research direction.

**Paper Topic And Main Contributions:**

A problem stemming from the currently dominant way of pretraining language models (LMs) on a data that was collected up to a certain point in time is that the resulting model quickly becomes out-of-date. Prior work has shown that temporal misalignment, i.e., the temporal distance between the training data cutoff and actual model usage correlates with task performance (of course, this applies only to temporally-sensitive tasks).

Within this context, the submission proposes a new auxiliary task, namely fact duration prediction, and shows that (1) models can be trained to predict the duration of facts expressed in text and (2) that such predictions can be used to improve the calibration of question answering models when dealing with temporally-sensitive facts, such as "Where will the next Summer Olympics be held?".

The basic idea is that one first uses the proposed method to predict the duration of fact like "The next Summer Olympics are going to be held in Athens" and then uses that prediction to judge whether there is any temporal misalignment, either with the model's training data (in a closed-book QA setting) or the corpus used for retrieval (in an open-book QA setting). If deemed a temporal misalignment, there are several options, such as reducing the confidence of model predictions post-hoc, or dynamically selecting wether to answer a question closed-book (i.e., no retrieval) or open-book (retrieval). Results show that the prediction of temporal misalignment is accurate enough to improve model calibration when used in this way.

**Reasons To Accept:**

- The problem the submission tackles, i.e., temporal misalignment between LM training and use, is highly relevant. Of course, one paper alone will not solve this problem, but I think it makes a valuable contribution.

- The proposed task, fact duration prediction, is well-defined, feasible, and looks like it could have other uses besides those described in the submission.

- While experiments on more than the two models used in the submission would have made a stronger case, the experiments look like a solid proof-of-concept. The different ways of using the temporal misalignment prediction for improving calibration (showcased in the second half of the paper) appear very useful.

- The submission is well-written overall and easy to understand.

**Reasons To Reject:**

- none

**Reproducibility:**

4: Could mostly reproduce the results, but there may be some variation because of sample variance or minor variations in their interpretation of the protocol or method.

**Reviewer Confidence:**

4: Quite sure. I tried to check the important points carefully. It's unlikely, though conceivable, that I missed something that should affect my ratings.

**Typos Grammar Style And Presentation Improvements:**

Table 5 should include scores for systems without adaptive inference (if I understand correctlly, that would be the EM scores in Table 3) so that the reader doesn't have to backtrack in order to understand what is being refered by "either system on its own" (lines 529-530):

> We report our results in the first row of Table 5, which shows that this outperforms either system on its own

---

> ### Author Rebuttal · Authors · 2023-08-28
>
> Thank you for your thoughtful review! We agree that the edits you suggested for Table 5 would clarify the comparisons made in our work. We plan to include these edits in our revisions.

---

### Official Review · Reviewer_BbjM · 2023-08-04

**Soundness:** 4

**Excitement:**

3: Ambivalent: It has merits (e.g., it reports state-of-the-art results, the idea is nice), but there are key weaknesses (e.g., it describes incremental work), and it can significantly benefit from another round of revision. However, I won't object to accepting it if my co-reviewers champion it.

**Paper Topic And Main Contributions:**

This paper proposes a method to deal with temporal misalignment in open-retrieval QA systems. To do so, a "fact duration prediction" task is proposed, where a model is trained to predict the lifespan of facts. The model is trained using large-scale distant supervision, either as a classification or regression. The trained model then can be used to aid the calibration of QA models by rescaling model confidence using the fact duration error measured by the duration prediction model.


**Questions For The Authors:**

- If possible, it would be interesting to see whether LLMs have solved the calibration issue or are simply abstaining.
- I'm curious about the relationship to the parallel question of mathematical reasoning, as fact duration prediction could be viewed as a rudimentary type of mathematical question. It would be informative to include results with math QA models.

**Reasons To Accept:**

This is an interesting and somewhat overlooked problem in pretrained language models. The distantly supervised model shows moderate performance in fact duration prediction, showing there is a large room for improvement, which can encourage more research in this area.
The paper's exploration of temporal degradation in relation to QA model calibration should be commended.

**Reasons To Reject:**

The scope of the problem is limited, and while LLMs do not entirely resolve the problem, LLMs are able to abstain from answers with potential temporal misalignment.

**Reproducibility:**

4: Could mostly reproduce the results, but there may be some variation because of sample variance or minor variations in their interpretation of the protocol or method.

**Reviewer Confidence:**

2: Willing to defend my evaluation, but it is fairly likely that I missed some details, didn't understand some central points, or can't be sure about the novelty of the work.

---

> ### Author Rebuttal · Authors · 2023-08-28
>
> Thank you for your careful review!
>
>
> Regarding your comment about whether existing LLMs have already solved the issue of temporal misalignment:
>
> > We noticed newer proprietary LLMs (e.g., ChatGPT) are equipped with an ability to abstain from answering with rationale of temporally dependent questions. Yet, from our investigation (L634-653) their ability is fairly brittle and the problem remains unresolved. Specifically, we find that LLMs tend to over-abstain from answering, abstaining from 79% of questions whose answers did not change between 2018 and 2021 (and 95% of temporally dependent questions).
>
> > Furthermore, unlike our systems, they do not provide an estimate of how likely it is that its predicted answer is out-of-date. This piece of information can be important for users deciding whether or not to trust the given answer.
>
> > Lastly, as far as we are aware, none of the open-sourced LLM show such abstaining functionality, adding value to our study.
>
>
> Regarding connections to mathematical reasoning:
>
> > Great point! We have also noted there is a relationship between fact duration prediction and other quantitative reasoning tasks (L556-571). We did not include systems designed for mathematical-reasoning tasks (e.g., DROP) as baselines as these systems are designed to perform complex reasoning using only the values given within the context. Fact duration prediction, however, is a task that only requires very simple mathematical reasoning using values that are recalled from real-world facts and commonsense knowledge(e.g., the US president is re-elected every 4 years, news anchors usually last many years). Therefore, the challenges these two tasks pose and the methods developed for each are quite orthogonal. We will add this discussion to the final draft of the paper.

---

### Official Review · Reviewer_Cm26 · 2023-08-11

**Soundness:** 4

**Excitement:**

4: Strong: This paper deepens the understanding of some phenomenon or lowers the barriers to an existing research direction.

**Paper Topic And Main Contributions:**

The paper addresses the problem of temporal misalignment in a QA setting. Language models can be trained to answer questions based on their pre-training data. However, this knowledge becomes outdated as time passes. The paper proposes predicting for how long would a fact remain true to mitigate temporal misalignment. By estimating when a fact would "expire", the models can be made less confident about their predictions. The paper introduces two fact duration prediction systems based on BERT and evaluate these. Furthermore, the paper combine these systems with QA-models with goal of calibrating their predictions (making them less confident about outdated answers). Additionally, the paper uses fact duration prediction to combine open- and closed-book QA system, falling back on the open-book system, when the closed-book system provides outdated answers. The benefit of the proposed system is also shown in re-ranking answers based on two corpora with different dates in an open-QA system.

**Questions For The Authors:**

Question Cm26_A: In the last experiment ( Two Corpora: Relevancy vs. Recency): Why would one want to perform inference based on two corpora (and double the inference costs), while using only one corpus (the most recent) gives better results ?
Question Cm26_B: L447-452: how is this calibrator used with the fact duration system?

**Reasons To Accept:**

* The paper addresses an interesting problem from a new perspective
* The proposed solution can help make QA-systems more robust by making them more "aware" of the validity of their answers with respect to time.
*  Several scenarios are considered in the evaluation

**Reasons To Reject:**

NA

**Reproducibility:**

3: Could reproduce the results with some difficulty. The settings of parameters are underspecified or subjectively determined; the training/evaluation data are not widely available.

**Reviewer Confidence:**

3: Pretty sure, but there's a chance I missed something. Although I have a good feel for this area in general, I did not carefully check the paper's details, e.g., the math, experimental design, or novelty.

**Typos Grammar Style And Presentation Improvements:**

* It is not clear what is the idea behind ECE
* It is confusing to look at Table 1, when starting to read the paper
* Table 5: having the baseline you compare against in the same table would make the results easier to compare
* L 268: them explore: "to" missing
* L286: ... 70 different: "of" missing
* L364: reflect the: "on" missing
* L447-448: predict predicts

---

> ### Author Rebuttal · Authors · 2023-08-28
>
> Thank you for your insightful comments and notes on presentation! We plan to incorporate your suggestions in our revisions. Regarding your specific questions, we respond to each below:
>
> Question A: “In the last experiment (Two Corpora: Relevancy vs. Recency): Why would one want to perform inference based on two corpora (and double the inference costs), while using only one corpus (the most recent) gives better results?”
>
> > We agree that in the ideal scenario for retrieval would be to have one corpus where each article is constantly updated and outdated information is removed. This, however, is not the case for most corpera outside of Wikipedia. News articles, for instance, are usually written once and are never edited after they are published. News corpera contain articles written over a large period of time, where older articles contain a mix of correct facts, which have not changed, and outdated facts, which newer articles have updated. In using news articles as a knowledge corpus, we cannot completely throw out old articles without losing valuable information.
> This experiment simulates a setting when we have two retrieval corpus, each from a different time period. In this setting, only using the most recent corpus is akin to the oracle setting, where we are able to remove all facts from the older corpus that have been restated or updated in future articles. Here, we demonstrate that relying on fact duration prediction to determine when facts from an older article can be trusted recovers similar performance to this oracle. We will clarify our motivations for this scenario in our writing.
>
>
> Question B: “L447-452: how is this calibrator used with the fact duration system?”
>
> > This process is described in the equations on L425 and L430. In these equations, we use the calibrator described on L447-452 to get the confidence score, c(q, a). This confidence score can be interpreted as predicting “the likelihood that this answer is true as of 2018”. We then use our fact duration predictions (which are made independently from a separate model) to adjust this score based on the temporal misalignment. We use this adjusted confidence estimate as our misalignment-aware confidence score.

---

### Meta-Review · Area_Chair_KJwZ · 2023-09-19

**Recommendation:** 5

**Metareview:**

This paper addresses an important question that when an LLM is trained with outdated information, it fails to generate correct responses for time-sensitive questions. The paper introduces a factual direction prediction component to determine whether there is a temporal mismatch between the training data and test data. All reviewers appreciated the clarity of the problem space and solution, and solid experiments.

---

### Decision · Program_Chairs · 2023-10-07

**Decision:**

Accept-Main

**Comment:**

This paper addresses an important question that when an LLM is trained with outdated information, it fails to generate correct responses for time-sensitive questions. The paper introduces a factual direction prediction component to determine whether there is a temporal mismatch between the training data and test data. All reviewers appreciated the clarity of the problem space and solution, and solid experiments.